# Multi-Modal Representation Learning via Semi-Supervised Rate Reduction for Generalized Category Discovery

## Abstract

Generalized Category Discovery (GCD) aims to identify both known and unknown categories, with only partial labels given for the known categories, posing a challenging open-set recognition problem. Recently, Visual-Language Models (VLMs) are employed to learn multi-modality representations for GCD task. Usually the representation learning approaches for multi-modal GCD are depend upon modality alignment. However, there is a lack of sufficient investigation on the underlying structure of distributions. In this paper, we propose a novel and effective multi-modal representation learning framework for GCD via Semi-Supervised Rate Reduction, called $SSR^2$-GCD, which is able to learn cross-modality representations with desired structural properties to align the intra-modality relationships. Moreover, we also integrate semantic information from prompt candidates by leveraging the inter-modal alignment offered by VLMs. Experiments conducted on generic and fine-grained benchmark datasets demonstrate the superior performance of our approach.

## 1 Introduction

Generalized Category Discovery (GCD) has emerged as a natural and challenging extension of open-set recognition with the aim of discovering categories (i.e., patterns) in the open world (Vaze et al., 2022; Scheirer et al., 2012). The goal of GCD is to recognize both known and unknown categories, going beyond the standard open-set recognition problem by leveraging knowledge learned from known categories to discover unknown categories. For example, in the typical setting of GCD task, half of categories are partially labeled (known) and the rest of categories remain unlabeled (unknown). This setting is relevant to real-world exploration scenarios in which data exhibit a mixture of known and unknown class structures.

Existing approaches to address the GCD problem usually follow a two-phases framework: a) generating representations for images by fine-tuning the pre-trained models, and b) applying clustering algorithms on the learned representations of all unlabeled data. However, these methods lack of effective signals to transfer knowledge from known categories to discovering unknown categories, e.g., using the given partial labels to improve the performance on known categories but recovering the unknown categories in unsupervised or self-supervised manner.

For human being, it is consciously or unconsciously leveraging information with multiple sources of cues from known categories to recognize unknown categories. Recently, there are a few attempts to explore multi-modal frameworks for GCD by integrating information from textual modality. For example, CLIP-GCD (Ouldnoughi et al., 2023) leverages a knowledge database to search for similar texts of query images, TextGCD (Zheng et al., 2025) constructs prompts from tag and attribute lexicons, and GET (Wang et al., 2025) learns a textual inversion network to generate prompts. Previous multi-modal GCD frameworks have explored introducing textual cues into visual datasets, yet lack sufficient investigation on the underlying structure of the distribution in multi-modal representation learning. For example, these frameworks simply incorporate existing CLIP-style inter-modality loss (Radford et al., 2021) or GCD-style intra-modality loss (Vaze et al., 2022) for representation learning. They neither offer clear insights into the role of inter-modal and intra-modal interactions, nor address the inherent issues in existing representation learning paradigms.

In this paper, we propose a novel framework, called Semi-Supervised Rate Reduction for Generalized Category Discovery ($SSR^2$-GCD), to address multi-modality representation learning in the GCD task. To be specific, we incorporate a Semi-Supervised Rate Reduction ($SSR^2$) principle to learn structured representations, in which the consistency of intra-modal representations is encouraged and the representations across known and unknown categories are evenly compressed. Unlike the existing multi-modal GCD methods that presuppose the criticality of the inter-modal alignment, we find empirically that the inter-modal alignment is not that important offering novel insights. This finding is thoroughly discussed and validated through extensive experiments. Moreover, we also present a Retrieval-based Text Aggregation (RTA) strategy to enhance the text generation, in which the information from a larger amount of prompt candidates is integrated to generate semantic-rich textual representations.

**Contribution.** The main contributions of the paper are highlighted as follows.

1. We incorporate a Semi-Supervised Rate Reduction principle for Generalized Category Discovery ($SSR^2$-GCD) to learn structured representations by which both the known and unknown categories are evenly compressed.

2. We demonstrate that inter-modal alignment can be non-essential, offering insights for representation learning in multi-modal GCD frameworks.

3. We conduct extensive experiments on eight datasets, showing superior performance of the proposed approach.

## 2 RELATE WORK

**Generalized Category Discovery (GCD).** GCD considers a realistic scenario in which the unlabeled dataset includes samples from both known and unknown categories, requiring simultaneous discovering of known and unknown categories. Vaze et al. (2022) first address the GCD problem by leveraging supervised and self-supervised contrastive learning to refine features produced by pretrained vision models, and clustering via semi-supervised $k$-means algorithm. Then, a number of methods for GCD follow such a pipeline. For instance, SimGCD (Wen et al., 2023), which is a notable baseline for GCD, introduces a parametric classifier to replace the non-parametric clustering and incorporates an entropy regularization to alleviate the degradation of classifier on unknown categories; SelEx (Rastegar et al., 2024) leverages a hierarchical semi-supervised $k$-means and achieves better results on fine-grained datasets; GPC (Zhao et al., 2023) employs Gaussian mixture models to learn representations while simultaneously estimating the number of unknown categories; PromptCAL (Zhang et al., 2023) introduces a contrastive affinity learning framework, in which auxiliary visual prompts are incorporated to address the false negative-induced category collision issue; SPTNet (Wang et al., 2024a) proposes a spatial prompt tuning method that iteratively finetunes the backbone and learns pixel-level prompts, effectively transferring semantic knowledge in GCD; HypCD (Liu et al., 2025) introduces a framework that considers both hyperbolic distance and the angle between samples to learn hierarchy-aware representations. However, all these methods mentioned above exploit the visual cues. Note that for human being, each known visual category inherently corresponds to specific semantic meanings described by natural language and thus it is typically incorporating cues from multiple modalities to recognize new categories.

**Multi-modal Generalized Category Discovery.** Vision-language pre-trained models (VLMs) such as CLIP (Radford et al., 2021) embed images and text into an aligned semantic space by pulling the representations of positive image-text pairs, enabling various downstream applications. Recently, multi-modal GCD methods (Su et al., 2024; Ouldnoughi et al., 2023; Zheng et al., 2025; Wang et al., 2025) leverage the external guidance of textual modality brought by VLMs to facilitate knowledge transfer between known and unknown categories. For instance, CLIP-GCD (Ouldnoughi et al., 2023) leverages a knowledge database to generate texture descriptions and concatenates both the visual embedding and text embedding obtained from a frozen CLIP backbone for clustering; MM-GCD (Su et al., 2024) propose a multi-modal framework to align with both the feature and output spaces of different modalities using contrastive learning and distillation technique; TextGCD (Zheng et al., 2025) proposes a retrieval-based text generation method to generate semantic-rich texture descriptions by incorporating abundant tag and attribute candidates, and introduces a co-teaching technique to align the clustering outputs of vision and text branches. However, TextGCD simply

employs the inter-modal contrastive loss of CLIP to fine-tune the backbone, while using the similarities within each modality for intra-modal clustering. More recently, GET (Wang et al., 2025) trains a textual inversion network (Baldrati et al., 2023) that maps the image embedding to pseudo textual token for unlabeled images. To alleviate the distributional shift of the text manifold towards the image manifold, GET aligns the embedding of pseudo-token with the embedding of class names on labeled data. However, only class names of known categories are available in the GCD setting, which prevents the textual inversion network from generating high-quality pseudo-tokens for images belonging to unknown categories. While existing multi-modal GCD approaches have achieved promising results, they fail to fully exploit the information provided by the textual modality.

## 3 PRELIMINARIES

### 3.1 PROBLEM NOTATION

Denote the data set as $\mathcal{D}_L \cup \mathcal{D}_U$, which consists of labeled data $\mathcal{D}_L = (\boldsymbol{x}_i, y_i^l)_{i=1}^M \subseteq \mathcal{X} \times \mathcal{Y}_L$ and unlabeled data $\mathcal{D}_U = (\boldsymbol{x}_i, y_i^u)_{i=1}^N \subseteq \mathcal{X} \times \mathcal{Y}_U$, where $\mathcal{Y}_L$ and $\mathcal{Y}_U$ denote the label spaces, and $\mathcal{Y}_L \subset \mathcal{Y}_U$. In the GCD setting, the labeled samples in $\mathcal{D}_L$ are from the known categories; whereas unlabeled samples in $\mathcal{D}_U$ are either from the known categories nor from some unknown categories. As usual, the total number of categories $K = |\mathcal{Y}_U|$ is given or can be estimated. The goal of GCD is to estimate the labels of samples in $\mathcal{D}_U$.

### 3.2 MULTI-MODAL GCD BASELINES

In this section, we review the multi-modal GCD baselines relevant to our approach. The common practice of multi-modal GCD frameworks consist of three components, i.e., text generation, representation learning, and clustering.

**Text Generation.** To introduce textual cues to visual datasets, inherent inter-modal alignment capability of pre-trained VLMs is leveraged to generate pseudo-texts for query images. Among these approaches, retrieval-based methods (Li et al., 2024; Zheng et al., 2025) construct a prompt database $\mathcal{P}$ and search for the optimal prompt $p \in \mathcal{P}$ that maximizes the cosine similarity between the embeddings of query image and prompt candidates. For instance, TextGCD (Zheng et al., 2025) uses the class names in ImageNet (Deng et al., 2009) to construct the tag lexicon and leverages GPT3 (Brown et al., 2020) to generate distinguishing attributes for these tags. For each query image, the top-3 similar tags and top-2 similar attributes are used to construct the prompt: "most likely $\{tag_1\}$, perhaps $\{tag_2\}$, likely $\{tag_3\}$, most likely $\{attr_1\}$, perhaps $\{attr_2\}$".

**Representation Learning.** Given query images and their corresponding pseudo-texts, multi-modal GCD frameworks usually refine their representations simultaneously. For instance, TextGCD learns the representations by simply following the CLIP-style inter-modal contrastive loss, i.e.,

$$\mathcal{L}_{\text{CLIP}} = -\frac{1}{|B|} \sum_{i \in B} \log \frac{\exp\left(\boldsymbol{z}_i^{\text{I}\top} \boldsymbol{z}_i^{\text{T}}/\tau_c\right)}{\sum_{j \neq i} \exp\left(\boldsymbol{z}_i^{\text{I}\top} \boldsymbol{z}_j^{\text{T}}/\tau_c\right)}, \tag{1}$$

where $B$ denotes the mini-batch of data, $\boldsymbol{z}_i^{\text{I}}, \boldsymbol{z}_i^{\text{T}} \in \mathbb{R}^d$ denote the embeddings of $i$-th image and pseudo-text, and $\tau_c$ is the temperature factor. Optimizing $\mathcal{L}_{\text{CLIP}}$ encourages the inter-modal alignment between image and text manifolds, but does not account for intra-modal alignment within each modality. GET also encourages the inter-modal alignment in representation learning, while incorporating the widely-used supervised and unsupervised contrastive losses to align intra-modal relationships within each modality, i.e.,

$$\mathcal{L}_{\text{con}} = \lambda \mathcal{L}_{\text{con}}^{\text{s}} + (1 - \lambda)\mathcal{L}_{\text{con}}^{\text{u}},$$

$$\mathcal{L}_{\text{con}}^{\text{s}} = -\sum_{i \in B_l} \frac{1}{|\mathcal{N}_i|} \sum_{j \in \mathcal{N}_i} \log \frac{\exp\left(\boldsymbol{z}_i^\top \boldsymbol{z}_j'/\tau_a\right)}{\sum_{m \neq i} \exp\left(\boldsymbol{z}_i^\top \boldsymbol{z}_m'/\tau_a\right)}, \tag{2}$$

$$\mathcal{L}_{\text{con}}^{\text{u}} = -\sum_{i \in B} \log \frac{\exp\left(\boldsymbol{z}_i^\top \boldsymbol{z}_i'/\tau_b\right)}{\sum_{m \neq i} \exp\left(\boldsymbol{z}_i^\top \boldsymbol{z}_m'/\tau_b\right)},$$

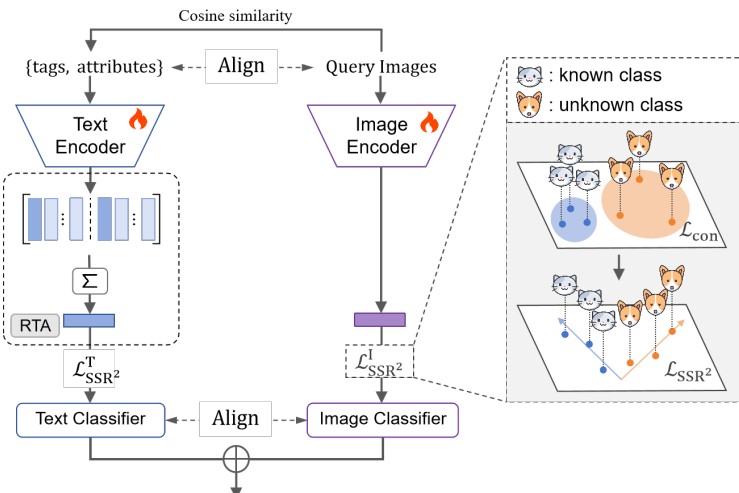

Figure 1: Illustration of Our Proposed Framework.

where $\lambda$ is the balancing parameter, $B_\ell$ denotes the labeled subset of $B$, $z_i$ and $z'_i$ are embeddings of augmented data pairs, $\mathcal{N}_i$ is data indices with the same label as $i$-th data, and $\tau_a$ and $\tau_b$ are the temperature parameters. Still, the intra-modal representation learning in multi-modal GCD frameworks adheres to the paradigm of uni-modal counterparts, and fails to resolve the inherent problems of this paradigm. That is, optimizing $\mathcal{L}_{\text{con}}$ results in the **imbalanced compression** of embeddings, since $\mathcal{L}_{\text{con}}^{\text{u}}$ pulls positive augmented data pairs across all categories, while $\mathcal{L}_{\text{con}}^{\text{s}}$ further pulls labeled data together only for known categories. As illustrated in Figure 1, such an imbalanced compression issue prevents clustering algorithms from accurately identifying cluster boundaries.

**Clustering.** Existing multi-modal GCD frameworks usually follow SimGCD (Wen et al., 2023) for clustering unlabeled data, which consists of a supervised cross-entropy loss (Krizhevsky et al., 2012) and an unsupervised self-distillation loss (Assran et al., 2022), i.e.,

$$\mathcal{L}_{\text{cls}} = \sum_{i \in B_l} \ell_{\text{CE}}(\boldsymbol{y}_i^*, \boldsymbol{y}_i) + \gamma \sum_{i \in B} \ell_{\text{CE}}(\boldsymbol{y}_i', \boldsymbol{y}_i) - \mu H(\overline{\boldsymbol{y}}), \tag{3}$$

where $\gamma$ and $\mu$ are the balancing parameters, $\ell_{\text{CE}}$ denotes the cross-entropy loss, $\boldsymbol{y}_i^*$ is the ground-truth label of the $i$-th image, $\hat{\boldsymbol{y}}_i$ is the prediction of the $i$-th embedding $z_i$, $\boldsymbol{y}_i'$ is the prediction of the augmented counterpart $z'_i$ with a sharper temperature in `Softmax`. The mean entropy regularizer (Van Gansbeke et al., 2020) $H(\overline{\boldsymbol{y}}) = -\sum_k \overline{\boldsymbol{y}}^{(k)} \log \overline{\boldsymbol{y}}^{(k)}$ is introduced to prevent degenerated predictions in new categories, where $\overline{\boldsymbol{y}} = \frac{1}{2|B|} \sum_{i \in B} (\boldsymbol{y}_i + \boldsymbol{y}_i')$ denotes the mean prediction of $\boldsymbol{y}_i + \boldsymbol{y}_i'$ in a mini-batch using the same temperature, and $\overline{\boldsymbol{y}}^{(k)}$ is the value of $\overline{\boldsymbol{y}}$ in the $k$-th class. For example, GET trains a single MLP by optimizing the loss in Eq.(3) to produce predictions for both image and text embeddings, while TextGCD implements dual-branch classifiers to handle image and text embeddings, respectively. Additionally, TextGCD uses the co-teaching strategy, in which high-confidence samples are used to supervise the learning of classifiers, i.e.,

$$\mathcal{L}_{\text{co-teach}} = \sum_{i \in \mathcal{S}^{\text{I}}} \ell_{\text{CE}}(\boldsymbol{y}_i^{\text{I}}, \boldsymbol{y}_i^{\text{T}}) + \sum_{j \in \mathcal{S}^{\text{T}}} \ell_{\text{CE}}(\boldsymbol{y}_j^{\text{T}}, \boldsymbol{y}_j^{\text{I}}), \tag{4}$$

where $\mathcal{S}^{\text{I}}, \mathcal{S}^{\text{T}}$ are the sample sets selected based on the confidence score of $\boldsymbol{y}^{\text{I}}$ and $\boldsymbol{y}^{\text{T}}$, respectively.

# 4 OUR PROPOSED APPROACH: SSR$^2$-GCD

Roughly, our proposed framework consists of three modules: a) a Retrieval-based Text Aggregation (RTA) strategy for text generation; b) a Semi-Supervised Rate Reduction (SSR$^2$) principle for representation learning; and c) dual-branch classifiers to learn pseudo-labels from each modality. Specifically, at first we use RTA to aggregate embeddings of prompts to incorporate textual information that is helpful in discovering unknown categories; then, we use the Semi-Supervised Rate

Reduction ($SSR^2$) objective to learn structured representations with desired properties. The dual-branch classifiers are deployed to learn pseudo-labels from each modality. For clarity, we illustrate our proposed framework in Figure 1.

## 4.1 RETRIEVAL-BASED TEXT AGGREGATION

For text generation, we adopt TextGCD (Zheng et al., 2025) to construct tag and attribute lexicons, and then search for similar tags and attributes, since we find that incorporating more information from multiple tag and attribute candidates is helpful in discovering patterns of unknown categories. Still, due to CLIP's limitation in handling long textual prompts, the way of constructing prompts in TextGCD is sub-optimal, because CLIP fails to generate satisfactory embeddings for prompts exceeding 20 tokens (Zhang et al., 2024). Given CLIP's text encoder $\mathcal{F}^T$ and tokenizer $\mathcal{T}$, we use the text encoder to embed tag and attribute prompts, and then compute the textual embedding by:

$$\boldsymbol{z}^T = \sum_{i=1}^{c} \sigma_i \mathcal{F}^T(\mathcal{T}(a_i)) + \sum_{i=1}^{c} \sigma_i \mathcal{F}^T(\mathcal{T}(b_i)), \tag{5}$$

where $a_i$ and $b_i$ are the $i$-th similar tags and attributes, respectively, as ranked by the cosine similarity between their embeddings and the embedding of query image $z^I$, $c$ means that only the top-$c$ most similar tags and attributes are considered, and the variable $\sigma_i$ assigns higher weights to the most-similar tag and attribute and aggregates information from other candidates, i.e.,

$$\sigma_i = \begin{cases} 1 - \alpha & \text{if } i = 1 \\ \frac{\alpha}{c-1} & \text{otherwise}, \end{cases} \tag{6}$$

where $\alpha > 0$ is hyper-parameter (e.g., $\alpha = 0.5$). This method aggregates richer information from a larger set of candidates (e.g., $c = 4$).

## 4.2 SEMI-SUPERVISED RATE REDUCTION PRINCIPLE FOR REPRESENTATION LEARNING

In this section, we tackle the imbalanced compression issue in existing representation learning methods. Specifically, we propose a **S**emi-supervised **R**ate **R**eduction (**SSR**$^2$) approach to to learn structured representations from intra-modal relationships while achieving even compression across known and unknown categories. Inspired by the principle of Maximal Coding Rate Reduction (Yu et al., 2020), a structured representation learning technique originally developed for supervised settings, we formulate SSR$^2$ objective as follows:

$$\mathcal{L}_{SSR^2} = -R(\mathbf{Z}) + R_c^s(\mathbf{Z}, \mathbf{Y}^*) + R_c^u(\mathbf{Z}, \mathbf{Y}), \tag{7}$$

where

$$R(\mathbf{Z}) := \log \det \left( \mathbf{I} + \frac{d}{N\epsilon^2} \mathbf{Z}\mathbf{Z}^\top \right),$$

$$R_c^s(\mathbf{Z}_s, \mathbf{Y}^*) := \frac{1}{N} \sum_{j=1}^{k} \log \det \left( \mathbf{I} + \frac{d}{N_j^s \epsilon^2} \mathbf{Z}_s \text{Diag}(\mathbf{Y}_j^*) \mathbf{Z}_s^\top \right),$$

$$R_c^u(\mathbf{Z}_u, \mathbf{Y}) := \frac{1}{N} \sum_{j=1}^{k} \log \det \left( \mathbf{I} + \frac{d}{N_j^u \epsilon^2} \mathbf{Z}_u \text{Diag}(\mathbf{Y}_j) \mathbf{Z}_u^\top \right),$$

where $\mathbf{Z}$ denotes the embeddings of a mini-batch, $\mathbf{Z}_s, \mathbf{Z}_u$ are the embeddings of labeled and unlabeled data, $\mathbf{Y}^*$ denotes ground-truth labels, $\mathbf{Y}$ denotes the pseudo-labels predicted by classifiers, $\mathbf{I}$ is the identity matrix, $k$ is the number of categories, $\epsilon > 0$ is the hyper-parameter ($\epsilon = 0.5$ in experiments), $N_j^s$ is the number of labeled data points assigned by $\mathbf{Y}^*$ that belong to the $j$-th class, and $N_j^u$ is the number of unlabeled data points assigned by $\mathbf{Y}$ that belong to the $j$-th class.

During training, we fine-tune the image and text encoders of CLIP (Radford et al., 2021), and the loss for each encoder is analogous to $\mathcal{L}_{SSR^2}$, with the replacement of embedding $\mathbf{Z}$ and pseudo-labels $\mathbf{Y}$ by image and text embeddings $\mathbf{Z}^I, \mathbf{Z}^T$ and pseudo-labels of dual-branch classifiers $\mathbf{Y}^I, \mathbf{Y}^T$, i.e.,

$$\begin{aligned} \mathcal{L}_{SSR^2}^I &= -R(\mathbf{Z}^I) + R_c^s(\mathbf{Z}^I, \mathbf{Y}^*) + R_c^u(\mathbf{Z}^I, \mathbf{Y}^I), \\ \mathcal{L}_{SSR^2}^T &= -R(\mathbf{Z}^T) + R_c^s(\mathbf{Z}^T, \mathbf{Y}^*) + R_c^u(\mathbf{Z}^I, \mathbf{Y}^T). \end{aligned} \tag{8}$$

The contributions of our proposed SSR$^2$ are twofold. On the one hand, maximizing the term $R(\cdot)$ in Eq. 7 expands the whole embeddings globally while minimizing the terms $R_c^{\text{s}}(\cdot)$ and $R_c^{\text{u}}(\cdot)$ encourages the embeddings of each category to span low-dimensional subspaces with even matrix ranks, as proved in (Yu et al., 2020; Wang et al., 2024b). Owing to such a desired property, the representations in each category are **evenly compressed**. Additionally, the proposed SSR$^2$ focuses on aligning intra-modal relationships and accommodate the discrepancies between modalities. We find that the inter-modal alignment leads to **intra-modal misalignment** and can be unnecessary in multi-modal GCD frameworks, as discussed and validated in our experiments. To our knowledge, this is the first work to address the imbalanced compression issue in contrastive-based representation learning, and to rethink the necessity of inter-modal alignment in the context of multi-modal GCD.

## 4.3 DUAL-BRANCH CLUSTERING

To fully discover the differences between modalities, we deploy dual-branch classifiers to tackle with image and text embeddings. The training of the two branch classifiers is conducted concurrently with the representation learning and is divided into warm-up and alignment stages.

**Warm-up Stage.** Following SimGCD (Wen et al., 2023), we adopt the loss $\mathcal{L}_{\text{cls}}$ in Eq. (3) for training classifiers. Specifically, the dual-branch classifiers accept image and text embeddings as the input, yielding modality-specific clustering losses $\mathcal{L}_{\text{cls}}^{\text{I}}$ and $\mathcal{L}_{\text{cls}}^{\text{T}}$. By combining the representation losses in Eq. (8), the total loss during the warm-up stage becomes:

$$\mathcal{L}_{\text{warm}} = \mathcal{L}_{\text{SSR}^2}^{\text{I}} + \mathcal{L}_{\text{SSR}^2}^{\text{T}} + \mathcal{L}_{\text{cls}}^{\text{I}} + \mathcal{L}_{\text{cls}}^{\text{I}}. \tag{9}$$

**Alignment Stage.** To align the orders of pseudo-labels predicted by dual-branch classifiers, we follow the co-teaching strategy of TextGCD (Zheng et al., 2025). By combining the loss $\mathcal{L}_{\text{co-teach}}$ in Eq 4, the total training loss in the alignment stage is formulated as:

$$\mathcal{L}_{\text{align}} = \mathcal{L}_{\text{SSR}^2}^{\text{I}} + \mathcal{L}_{\text{SSR}^2}^{\text{T}} + \mathcal{L}_{\text{cls}}^{\text{I}} + \mathcal{L}_{\text{cls}}^{\text{T}} + \mathcal{L}_{\text{co-teach}}. \tag{10}$$

After training, the predicted pseudo-label of the $i$-th image is calculated by $\arg\max(\boldsymbol{y}_i^{\text{I}} + \boldsymbol{y}_i^{\text{T}})$.

**Remark.** The clustering algorithm of our framework is the same as that in previous multi-modal GCD methods. However, the key distinction lies in representation learning. Specifically, TextGCD emphasizes only on the inter-modal alignment without incorporating the intra-modal constraints, and the learning of dual-branch classifiers is based on intra-modal similarities. As criticized in (Mistretta et al., 2025), using inter-modal alignment loss without intra-modal constraint will lead to intra-modal misalignment, i.e., the intra-modal similarities might not correspond to the actual pair-wise relationships, thereby degrading the performance of intra-modal clustering. In contrast, the intra-modal relationships are well aligned by using $\mathcal{L}_{\text{SSR}^2}$, and thus the dual-branch classifiers produce satisfactory results. In addition, the predictions produced by the dual-branch classifiers are also utilized as self-supervised signals to guide the joint representation learning, as shown in Eq. (7).

## 5 EXPERIMENTS

**Datasets.** We evaluate the performance of GCD methods on generic datasets, i.e., CIFAR-10/-100 (Krizhevsky et al., 2009), and fine-grained datasets, i.e., CUB-200-2011 (Wah et al., 2011), Stanford Cars (Krause et al., 2013), Oxford Pets (Parkhi et al., 2012) and Oxford 102 Flowers (Nilsback & Zisserman, 2008). Following the GCD protocol (Vaze et al., 2022), half of the samples from the known categories are selected to form the labeled dataset $\mathcal{D}_L$, while the remaining samples constitute the unlabeled dataset $\mathcal{D}_U$.

**Metrics.** Given pseudo-labels and ground-truths of $\mathcal{D}_U$, one can compute the clustering accuracy (ACC) using Hungarian matching algorithm (Kuhn, 1955). Following the GCD protocol, we report ACC on all categories ("All"), on known categories ("Old"), and on unknown categories ("New"), respectively. The average ACC over 3 trials is reported.

**Implementation Details.** In Retrieval-based Text Aggregation, we use the CLIP-H/14 (Radford et al., 2021) to search prompt candidates to ensure a fair comparison to TextGCD (Zheng et al., 2025). During training, we use the CLIP-B/16 encoders to produce text and image features. We report the performance of uni-modal counterparts using DINO (Caron et al., 2021) with ViT-B/16.

Table 1: The average ACC (%) on generic and fine-grained datasets. "†" denotes the reproduction of using CLIP backbone.

| | Method | CIFAR-10 | | | CIFAR-100 | | | CUB | | | Stanford Cars | | | Oxford Pets | | | Flowers102 | | |
|---|---|---|---|---|---|---|---|---|---|---|---|---|---|---|---|---|---|---|---|
| | | All | Old | New | All | Old | New | All | Old | New | All | Old | New | All | Old | New | All | Old | New |
| DINOv1 | GCD | 91.5 | 97.9 | 88.2 | 73.0 | 76.2 | 66.5 | 51.3 | 56.6 | 48.7 | 39.0 | 57.6 | 29.9 | 80.2 | 85.1 | 77.6 | 74.4 | 74.9 | 74.1 |
| | GPC | 92.2 | 98.2 | 89.1 | 77.9 | 85.0 | 63.0 | 55.4 | 58.2 | 53.1 | 42.8 | 59.2 | 32.8 | - | - | - | - | - | - |
| | SimGCD | 97.1 | 95.1 | 98.1 | 80.1 | 81.2 | 77.8 | 60.3 | 65.6 | 57.7 | 53.8 | 71.9 | 45.0 | 87.7 | 85.9 | 88.6 | 71.3 | 80.9 | 66.5 |
| | PromptCAL | 97.9 | 96.6 | 98.5 | 81.2 | 84.2 | 75.3 | 62.9 | 64.4 | 62.1 | 50.2 | 70.1 | 40.6 | - | - | - | - | - | - |
| | SPTNet | 97.3 | 95.0 | **98.6** | 81.3 | 84.3 | 75.6 | 65.8 | 68.8 | 65.1 | 59.0 | 79.2 | 49.3 | - | - | - | - | - | - |
| | SelEx | 95.9 | 98.1 | 94.8 | 82.3 | 85.3 | 76.3 | 73.6 | 75.3 | 72.8 | 58.5 | 75.6 | 50.3 | 92.5 | 91.9 | 92.8 | - | - | - |
| | Hyp-SelEx | 96.7 | 97.6 | 96.3 | 82.4 | 85.1 | 77.0 | **79.8** | 75.8 | **81.8** | 62.9 | 80.0 | 54.7 | - | - | - | - | - | - |
| CLIP | SimGCD† | 96.6 | 94.7 | 97.5 | 81.6 | 82.6 | 79.5 | 62.0 | 76.8 | 54.6 | 75.9 | 81.4 | 73.1 | 88.6 | 75.2 | 95.7 | 75.3 | 87.8 | 69.0 |
| | CLIP-GCD | 96.6 | 97.2 | 96.4 | 85.2 | 85.0 | 85.6 | - | - | - | 62.8 | 77.1 | 55.7 | 70.6 | 88.2 | 62.2 | 76.3 | 88.6 | 70.2 |
| | TextGCD | 98.2 | 98.0 | 98.6 | 85.7 | **86.3** | 84.6 | 76.6 | **80.6** | 74.7 | 86.1 | 91.8 | 83.9 | 93.7 | 93.2 | 94.0 | 87.2 | 90.7 | 85.4 |
| | GET | 97.2 | 94.6 | 98.5 | 82.1 | 85.5 | 75.5 | 77.0 | 78.1 | 76.4 | 78.5 | 86.8 | 74.5 | 91.1 | 89.7 | 92.4 | 85.5 | 90.8 | 81.3 |
| | **Ours** | **98.5** | **98.3** | **98.6** | **86.4** | 86.2 | **86.9** | 78.3 | 78.5 | 78.2 | **89.2** | **93.1** | **87.3** | **95.7** | **95.1** | **96.0** | **93.5** | **93.3** | **93.9** |

The classifier parameters are set following the default configurations as in SimGCD (Wen et al., 2023) and TextGCD (Zheng et al., 2025). More details are provided in the supplementary.

## 5.1 Performance on Benchmark Datasets

We compare the performance of $SSR^2$-GCD with recent uni-modal GCD methods, including GCD (Vaze et al., 2022), GPC (Zhao et al., 2023), SimGCD (Wen et al., 2023), PromptCAL (Zhang et al., 2023), SPTNet (Wang et al., 2024a), SelEx (Rastegar et al., 2024) and HypCD with the SelEx backbone (Liu et al., 2025), and multi-modal GCD methods, including CLIP-GCD (Ouldnoughi et al., 2023), TextGCD (Zheng et al., 2025) and GET (Wang et al., 2025). Since that GET did not provide results on Oxford Pets and Flowers102, we reproduce the results with the open-source code. As shown in Tables 1, our method consistently outperforms all other multi-modal counterparts on all tested datasets. We can also observe that our method decreases the accuracy gap between "Old" and "New" categories, and we will discuss it in the latter section. HypCD achieves the highest accuracy on CUB when using SelEx as the backbone. HypCD changes the embedding space from the Euclidean the hyperbolic, and thus it is complementary to our method. In addition, our method excels notably on the datasets Stanford Cars and Flowers102, achieving an accuracy of 89.2% and 93.5% on "All" categories, outperforming all other baselines by 3.1% and 6.3%, respectively. Note that CLIP performs poorly in out-of-domain datasets including Flowers102, achieving an accuracy of 70.4% of zero-shot classification (Radford et al., 2021). Our method effectively refines the representations generated by CLIP to the target domain of Flowers102 and yields satisfactory performance.

## 5.2 Evaluation on Representation Learning

**Is Inter-Modal Alignment Necessary?** To evaluate the effect of using inter-modal and intra-modal alignment in our framework, we keep the text generation and classification methods the same and report the performance of using different losses for representation learning, including the inter-modal loss (i.e., $\mathcal{L}_{\text{CLIP}}$ in Eq. (1)), the intra-modal losses (i.e., $\mathcal{L}_{\text{con}}$ in Eq. (2) and our $\mathcal{L}_{\text{SSR}^2}$ in Eq. (7)) and their combinations (e.g., $\mathcal{L}_{\text{CLIP}} + \mathcal{L}_{\text{con}}$). As can be seen in Table 2, we find that encouraging inter-modal alignment alone provides merely limited performance gain when compared to using the intra-modal losses alone, since that the learning of classifiers is based on the intra-modal relationships within each modality. Specifically, our framework achieves the highest accuracy on five benchmark datasets when using the proposed $\mathcal{L}_{\text{SSR}^2}$. Our framework trained with supervised and unsupervised contrastive loss $\mathcal{L}_{\text{con}}$ is still a strong intra-modal learning baseline, achieving the highest accuracy on CIFAR-100. Interestingly, combining $\mathcal{L}_{\text{CLIP}}$ with intra-modal losses $\mathcal{L}_{\text{con}}$ or $\mathcal{L}_{\text{SSR}^2}$ cannot significantly improve the clustering accuracy and even may obstacle the intra-modal learning. Specifically, "$\mathcal{L}_{\text{con}}$" outperforms "$\mathcal{L}_{\text{CLIP}} + \mathcal{L}_{\text{con}}$" on four datasets, except for CUB and Oxford Pets; whereas $\mathcal{L}_{\text{SSR}^2}$ surpasses "$\mathcal{L}_{\text{CLIP}} + \mathcal{L}_{\text{SSR}^2}$" on all datasets, suggesting that explicitly imposing inter-modal alignment might hinder the intra-modal representation learning.

**Discussions and Evaluations.** For the reason why the inter-modal alignment is no longer necessary, we account it in the hypothesis of clustering algorithms. On the one hand, the Retrieval-based Text

Table 2: Evaluation of different representation learning methods. Average ACC (%) on "All" categories is reported. "N/A" denotes using frozen CLIP.

| Rep. Losses | Inter | Intra | CIFAR-10 | CIFAR-100 | CUB | Stanford Cars | Oxford Pets | Flowers102 |
|---|---|---|---|---|---|---|---|---|
| N/A | ✗ | ✗ | 97.9 | 84.1 | 74.5 | 86.0 | 91.9 | 87.4 |
| $\mathcal{L}_{\text{CLIP}}$ | ✓ | ✗ | 98.3 | 86.0 | 76.7 | 87.0 | 94.1 | 89.7 |
| $\mathcal{L}_{\text{con}}$ | ✗ | ✓ | 98.4 | **86.7** | 77.5 | 87.9 | 94.9 | 91.8 |
| $\mathcal{L}_{\text{SSR}^2}$ | ✗ | ✓ | **98.5** | 86.4 | **78.3** | **89.2** | **95.7** | **93.5** |
| $\mathcal{L}_{\text{CLIP}}+\mathcal{L}_{\text{con}}$ | ✓ | ✓ | 98.2 | 86.3 | 78.0 | 86.7 | 95.0 | 90.9 |
| $\mathcal{L}_{\text{CLIP}}+\mathcal{L}_{\text{SSR}^2}$ | ✓ | ✓ | 98.3 | 86.1 | 77.2 | 88.1 | 95.0 | 92.9 |

(a) Visual modality     (b) Textual modality     (c) Image embeddings     (d) Text embeddings

Figure 2: $R_e$ curves with different losses on (a)-(b): Flowers102 and (c)-(d): Stanford Cars datasets.

Aggregation strategy and the clustering via dual-branch classifiers have fully utilized the consistency between modalities, while further encouraging inter-modal alignment during the finetuning of multimodal representations may eliminate the discrepancies between modalities. On the other hand, most GCD frameworks, including ours, utilize a self-distillation loss for clustering. As detailed in Eq. (3), the self-distillation loss, which minimizes the cross-entropy between the predictions of augmented data pairs from the same modality, is built on the assumption that the intra-modal embeddings of augmented data pairs should be close enough to be assigned to the same cluster. In other words, this loss rely on the hypothesis that only involves the consistency of the intra-modal embedding pairs. To quantize such intra-modal consistency, we construct the adjacency matrices $W$ for each modality, in which the weighted edges are the cosine similarities of the intra-modal embedding pairs, i.e., $W_{i,j} = \boldsymbol{z}_i^\top \boldsymbol{z}_j$. Given the ground-truth categories $\{\mathcal{C}_1, \ldots, \mathcal{C}_k\}$ indicated by their labels, one can quantize the consistency by computing the ratio of edges within each category to edges between categories, i.e.,

$$R_e = \frac{1}{k}\sum_{\ell=1}^{k} R_e(\mathcal{C}_\ell), \quad R_e(\mathcal{C}_\ell) := \frac{\sum_{i\in\mathcal{C}_\ell}\sum_{j\in\mathcal{C}_\ell} W_{i,j}}{\sum_{i\in\mathcal{C}_\ell}\sum_{j\notin\mathcal{C}_\ell} W_{i,j}}. \tag{11}$$

Apparently, a larger value of $R_e$ indicates a higher ratio of correct intra-category links to incorrect inter-category links, reflecting better consistency of intra-modal embeddings.

In Figure 2 we show the $R_e$ curves with different representation losses (i.e., $\mathcal{L}_{\text{CLIP}}$, $\mathcal{L}_{\text{SSR}^2}$ and $\mathcal{L}_{\text{CLIP}}+\mathcal{L}_{\text{SSR}^2}$) on Flowers102. As can be seen, our framework achieves higher $R_e$ by using $\mathcal{L}_{\text{SSR}^2}$ when compared to using $\mathcal{L}_{\text{CLIP}}$. While $\mathcal{L}_{\text{CLIP}}$ can implicitly enhance intra-class discriminability to some extent, it is insufficient to ensure intra-modal alignment and thus achieves less satisfactory clustering accuracy. This verifies that the inter-modal alignment without any intra-modal constraint leads to intra-modal misalignment. Moreover, we can see that the $R_e$ curve of using $\mathcal{L}_{\text{CLIP}} + \mathcal{L}_{\text{SSR}^2}$ rises in step with that of using $\mathcal{L}_{\text{SSR}^2}$ early in training, eventually converges toward the curve of using $\mathcal{L}_{\text{CLIP}}$. This verifies that the inter-modal alignment may damage the learning of intra-modal consistency.

**SSR$^2$ is a Compressor for Balanced Representation.** To evaluate the effectiveness of our proposed $\mathcal{L}_{\text{SSR}^2}$, we conduct experiments to compare it to the widely used contrastive loss $\mathcal{L}_{\text{con}}$. To quantify the uniformity of the compression, we compute the effective rank, which is defined as the number of leading singular values whose cumulative energy proportion reaches 95% of all singular values, of image embeddings per category, i.e., $\{\texttt{rank}(\mathbf{Z}^{(i)})\}_{i=1}^{k}$, where $\mathbf{Z}^{(i)}$ is the sub-matrix formed by embeddings from $i$-th ground-truth category. We train our framework via using $\mathcal{L}_{\text{con}}$ and $\mathcal{L}_{\text{SSR}^2}$, respectively, and plot the averaged ranks of "Old" categories and "New" categories, i.e., $\frac{1}{|\mathcal{Y}_l|}\sum_{i\in\mathcal{Y}_l}\texttt{rank}(\mathbf{Z}^{(i)})$ and $\frac{1}{|(\mathcal{Y}_u\backslash\mathcal{Y}_l)|}\sum_{j\in(\mathcal{Y}_u\backslash\mathcal{Y}_l)}\texttt{rank}(\mathbf{Z}^{(j)})$ on Flowers102 and Oxford Pets, respectively, and display the results in Figure 3. We can see that the average rank of the "Old" categories dramatically decreases during the training period when using the loss $\mathcal{L}_{\text{con}}$, which is much

lower than that of the "New" categories. Intuitively, using the loss $\mathcal{L}_{\mathrm{con}}$ will overly compress the embeddings toward the contrastive prototypes of the known categories, and thereby damages the accuracy of clustering those data points in the boundaries of the category manifolds. In contrast, when training with the loss $\mathcal{L}_{\mathrm{SSR}^2}$, the average ranks of the 'Old' and "New" categories are preserved well. This indicates that the embeddings of each category are compressed evenly, resulting in comparable clustering accuracy of 'Old' and "New" categories (see, e.g., Table 1).

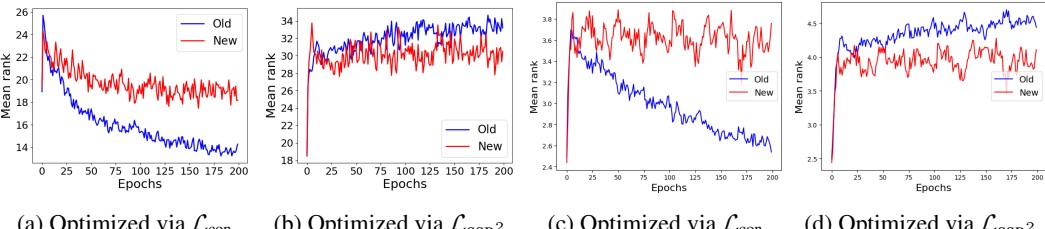

(a) Optimized via $\mathcal{L}_{\mathrm{con}}$    (b) Optimized via $\mathcal{L}_{\mathrm{SSR}^2}$    (c) Optimized via $\mathcal{L}_{\mathrm{con}}$    (d) Optimized via $\mathcal{L}_{\mathrm{SSR}^2}$

Figure 3: Effective ranks in (a)-(b): Oxford Pets, and (c)-(d): Flowers102 dataets.

## 5.3 ABLATION STUDY

To evaluate the effectiveness of each component in the proposed approach, we conduct a set of experiments on Stanford Cars and Flowers-102, in which the baseline is set up by using the most similar tag as pseudo-text, leveraging the frozen CLIP to generate embeddings, and clustering both image and text embeddings with shared parameters of a single classifier. Experimental results are listed in Table 3. We can read that the co-taught dual-branch classifiers outperform the baseline by identifying disparities between modalities. The proposed RTA integrates rich information from tag and attribute candidates, thereby enhancing clustering performance on "New" categories. The intra-modal representation learning with using the loss $\mathcal{L}_{\mathrm{SSR}^2}$ is also critical, as the training of classifiers relies on intra-modal relationships. Our framework achieves best performance by integrating all proposed components.

Table 3: Ablation study of different components.

| Dual | RTA | $\mathcal{L}_{\mathrm{SSR}^2}$ | Stanford Cars | | | Flowers102 | | |
|---|---|---|---|---|---|---|---|---|
| | | | All | Old | New | All | Old | New |
| × | × | × | 75.2 | 85.4 | 71.8 | 78.3 | 88.1 | 72.5 |
| ✓ | × | × | 81.7 | 90.3 | 77.1 | 83.9 | 88.3 | 81.3 |
| ✓ | ✓ | × | 86.0 | 91.0 | 83.4 | 87.4 | 90.8 | 85.5 |
| ✓ | × | ✓ | 85.5 | 91.7 | 82.2 | 89.1 | 91.8 | 88.0 |
| ✓ | ✓ | ✓ | **89.2** | **93.1** | **87.3** | **93.5** | **93.3** | **93.9** |

## 6 CONCLUSION

We have proposed a novel framework, called SSR$^2$-GCD, to tackle with the multi-modal GCD task. In particular, we incorporate a semi-supervised rate reduction principle to learn structured representations that are evenly compressed across categories, and we rethink the necessity of performing inter-modal alignment in multi-modal GCD framework. Moreover, we have presented a retrieval-based text aggregation approach to enhance text generation. We conducted extensive experiments on eight benchmark datasets and experimental results have verified our findings and demonstrated the effectiveness of our proposed SSR$^2$-GCD.

**Limitations.** The time and memory burden of our approach will increase when the number of candidates increases due to using multiple tag and attribute candidates as the input to CLIP encoders. Besides, in our framework currently the image and text cues are treated equally. A further exploration on the importance of each modality in the multi-modal GCD task is left for future work.

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

# SUPPLEMENTARY MATERIALS OF "MULTI-MODAL REPRESENTATION LEARNING VIA SEMI-SUPERVISED RATE REDUCTION FOR GENERALIZED CATEGORY DISCOVERY"

## A    ETHICS STATEMENT

This work adheres to the ICLR Code of Ethics. In this study, no human subjects or animal experimentation was involved. All datasets used were sourced in compliance with relevant usage guidelines, ensuring no violation of privacy. We have taken care to avoid any biases or discriminatory outcomes in our research process. No personally identifiable information was used and no experiments were conducted that could raise privacy or security concerns. We are committed to maintaining transparency and integrity throughout the research process.

## B    REPRODUCIBILITY STATEMENT

We have made every effort to ensure that the results presented in this paper are reproducible. All codes will be available to facilitate replication and verification. All datasets are publicly available, ensuring consistent and reproducible evaluation results. The experimental setup, including training steps, model configurations, and hardware details, is described in detail in appendix D to assist others in reproducing our experiments.

## C    LLM USAGE

In accordance with the ICLR 2026 policy, we confirm that no large language models (LLMs) were used at any stage of this work. All aspects of the research process—including problem formulation, methodology design, experimentation, analysis, and manuscript preparation—were carried out solely by the authors. The results, discussions, and conclusions presented in this paper are entirely based on the authors' own contributions, without reliance on any generative AI tools. The authors take full responsibility for the content under their names.

## D    EXPERIMENTAL DETAILS

### D.1    DATASETS

In Table D.1, we provide a statistical summarization of all generic and fine-grained datasets. Among these benchmarks, the generic datasets including CIFAR-10, CIFAR-100, ImageNet-100 and ImageNet-1k consist of categories of open-world, whereas the fine-grained benchmarks including CUB, Stanford Cars, Oxford Pets, and Flowers102 are largely domain-specific. Following the GCD protocol, we select the first 80 classes of CIFAR-100 as the known categories and use the first half of classes as the known categories for the other benchmarks.

Table D.1: Statistics of the benchmark datasets.

| Dataset | Labelled | | Unlabelled | |
|---|---|---|---|---|
| | #Image | #Class | #Image | #Class |
| CIFAR10 | 12.5K | 5 | 37.5K | 10 |
| CIFAR100 | 20.0K | 80 | 30.0K | 100 |
| ImageNet-100 | 31.9K | 50 | 95.3K | 100 |
| ImageNet-1K | 321K | 500 | 960K | 1000 |
| CUB | 1.5K | 100 | 4.5K | 200 |
| Stanford Cars | 2.0K | 98 | 6.1K | 196 |
| Oxford Pets | 1.9K | 19 | 5.5K | 37 |
| Flowers102 | 0.3K | 51 | 0.8K | 102 |

Table D.2: Model architectures. "*CLIP*" denotes the learnable parameters in CLIP and "*Cls*" denotes the classifiers.

| | |
|---|---|
| *CLIP* | Last residual attention block: $\mathbb{R}^{512} \rightarrow \mathbb{R}^{512}$ |
| | Image/text projector: $\mathbb{R}^{512} \rightarrow \mathbb{R}^{512}$ |
| *Cls.* | Linear: $\mathbb{R}^{512} \rightarrow \mathbb{R}^{k}$ |
| | Softmax |

Table D.3: **Pseudo-code** of the image augmentation.

```
from torchvision.transforms import *

Compose([
  RandomResizedCrop(32, BILINEAR)
  RandomHorizontalFlip(p=0.5),
  ColorJitter()
  ToTensor(),
  Normalize([0.485, 0.456, 0.406],[0.229, 0.224, 0.225])
  )])
```

## D.2 EXPERIMENT SETTINGS

**Model Architecture.** In Table D.2, we present the model architecture of the learnable parameters in the CLIP encoders and classifiers. The visual and textual branches share the same model architecture. Specifically, when using CLIP-B/16 as the backbone, we fine-tune the last residual attention block (which includes the multi-head self-attention mechanism, feed-forward network, and layer normalization), along with the image and text projectors of CLIP. Additionally, the dual-branch classifiers are learned with the fine-tuning of CLIP jointly.

**Data Preparation.** For each mini-batch data, we generate text embeddings for query images by integrating 4 similar tags and 4 similar attributes through the proposed retrieval-based text aggregation for all tested datasets. Then, both images and text are augmented into two views, and the augmentation strategies are the same across datasets, as detailed in Tables D.3 and D.4. The embeddings of augmented images and text are used for representation learning.

Table D.4: **Pseudo-code** of the text augmentation.

| **Text Augmentation Strategy** |
|---|
| **Input:** text |
| **For each** word **in** text: |
|    **If** len(word) $\geq$ 3: |
|      *index* $\leftarrow$ random(1, len(word)-2) |
|      *action* $\leftarrow$ random( {replace, delete, add, none} ) |
|      **Case** *action*: |
|        replace: word $\leftarrow$ replace random char at *index* |
|        delete: word $\leftarrow$ remove char at *index* |
|        add: word $\leftarrow$ insert random char at *index* |
|        none: **continue** |
| **Output:** augmented text |

**Training Settings.** During the training, we use the same setting for all tested datasets. We use the the stochastic gradient descent (SGD) with the momentum of $0.9$, the weight decay of $1 \times 10^{-4}$, and the cosine annealing learning rate decay for the training process. We train the model for 200 epochs and set the batch size to 128. The random seed for 3-trials is set to $[0, 1, 2]$. For representation learning, the initial learning rate for CLIP is set to $0.001$, and our proposed objective $\mathcal{L}_{\mathrm{SSR}^2}$ does not introduce an additional hyper-parameter. For clustering, the setting follows the default configurations as in SimGCD and TextGCD. Specifically, the initial learning rate for classifiers is set to $0.1$, the epochs

Table D.5: The mean $\pm$ std ACC (%) of TextGCD, GET and our approach on fine-grained datasets.

| Method | CUB | | | Stanford Cars | | | Oxford Pets | | | Flowers102 | | |
|---|---|---|---|---|---|---|---|---|---|---|---|---|
| | All | Old | New | All | Old | New | All | Old | New | All | Old | New |
| TextGCD | 76.6$\pm$0.6 | **80.6**$\pm$2.0 | 74.7$\pm$1.7 | 86.1$\pm$0.9 | 91.8$\pm$0.4 | 83.9$\pm$1.3 | 93.7$\pm$0.6 | 93.2$\pm$1.1 | 94.0$\pm$0.9 | 87.2$\pm$2.3 | 90.7$\pm$1.3 | 85.4$\pm$3.8 |
| GET | 77.0$\pm$0.5 | 78.1$\pm$1.6 | 76.4$\pm$1.2 | 78.5$\pm$1.3 | 86.8$\pm$1.5 | 74.5$\pm$2.2 | 91.1$\pm$1.0 | 89.7$\pm$1.6 | 92.4$\pm$1.2 | 85.5$\pm$0.5 | 90.8$\pm$1.5 | 81.3$\pm$1.7 |
| **Ours** | **78.3**$\pm$0.4 | 78.5$\pm$1.0 | **78.2**$\pm$0.9 | **89.2**$\pm$0.3 | **93.1**$\pm$0.9 | **87.3**$\pm$0.2 | **95.7**$\pm$0.1 | **95.1**$\pm$0.5 | **96.0**$\pm$0.4 | **93.5**$\pm$1.3 | **93.3**$\pm$1.8 | **93.9**$\pm$2.0 |

for warm-up stage is set to 10, and 60% of pseudo-labels of each categories are selected for co-teaching. All experiments are conducted on single NVIDIA GeForce RTX3090 GPU.

# E    MORE EXPERIMENTS

## E.1    EVALUATION ON MODEL STABILITY

In Tables D.5, we report the mean and standard deviation of accuracies over 3 trials, and compare them to those of the multi-modal counterparts TextGCD and GET. As can be seen, the performance of our proposed method is relatively stable.

## E.2    EVALUATION ON RETRIEVAL-BASED TEXT AGGREGATION

**Compared to other Text Generation Methods.**

We evaluate the effectiveness of proposed Retrieval-based Text Aggregation (RTA) and compare it to other text generation methods in multi-modal GCD. For a fair comparison, we replace the text generation methods in TextGCD (Zheng et al., 2025) and GET (Wang et al., 2025) with our RTA, marked the methods by adding a prefix 'RTA-'. Recall that we follow TextGCD and use CLIP with ViT-H/14 backbone to search prompt candidates from tag and attribute lexicons, we also report the performance of using different backbones. As can be seen from Table E.6 that, RTA significantly improves the performance of the GET framework as it incorporates more information from multiple prompts, which is helpful for discovering the patterns of unknown categories. Moreover, RTA also outperforms the text generation method proposed in TextGCD as it avoids the problem of CLIP in handling long texts and allows for aggregating more prompts. Finally, we also report the overall performance of the proposed $SSR^2$-GCD in Table E.7 by using different CLIP backbones to find candidates for the search of the candidates from the tag and attribute lexicons.

Table E.6: Evaluation of different text generation methods on benchmark datasets. Average ACC (%) on "All" categories is reported.

| Methods | Backbone | CIFAR-10 | CIFAR-100 | CUB | Stanford Cars | Oxford Pets | Flowers102 |
|---|---|---|---|---|---|---|---|
| TextGCD | ViT-H/14 | 98.2 | 85.7 | 76.6 | 86.1 | 93.7 | 87.2 |
| RTA-TextGCD | ViT-H/14 | 98.3 | 86.0 | 76.7 | 87.0 | 94.1 | 89.7 |
| TextGCD | ViT-B/16 | 97.1 | 83.0 | 73.2 | 83.0 | 90.9 | 83.1 |
| RTA-TextGCD | ViT-B/16 | 97.7 | 84.5 | 74.9 | 85.1 | 93.2 | 88.6 |
| GET | ViT-B/16 | 97.2 | 82.1 | 77.0 | 78.5 | 91.1 | 85.5 |
| RTA-GET | ViT-B/16 | 97.9 | 85.0 | 77.5 | 84.8 | 93.0 | 89.1 |

**Effect of Hyper-Parameter $\alpha$.** In our proposed Retrieval-based Text Aggregation (RTA), the hyper-parameter $\alpha$ serves as the balance factor of integrating most-similar candidates and other candidates. To evaluate the effect of hyper-parameter $\alpha$, we report the accuracy of "All" categories on 4 generic and fine-grained datasets, with varying values of $\alpha$ in $\{0.2, 0.3, 0.4, 0.5, 0.6, 0.7, 0.8\}$ while keeping other components in our framework unchanged. As shown in Table E.8, we observe that using the proposed RTA achieves its best performance on CIFAR-10, Stanford Cars and Flowers102 with $\alpha = 0.5$, and achieves its best performance on CIFAR-100 with $\alpha = 0.4$.

**Effect of the Number of Candidates.** In RTA, we select multiple candidates for tags and attributes. To evaluate the effect of the number of candidates, we report the accuracy of "All" categories on the generic and fine-grained datasets, with varying number of tags and attributes in $\{1, 2, 3, 4, 5\}$, respectively, while keeping other components in our framework unchanged. As shown in Figure E.1,

Table E.7: The average ACC (%) of our proposed SSR$^2$-GCD with different backbones for searching prompt candidates on generic and fine-grained datasets.

| Backbone | CIFAR-10 | | | CIFAR-100 | | | CUB | | | Stanford Cars | | | Oxford Pets | | | Flowers102 | | |
| --- | --- | --- | --- | --- | --- | --- | --- | --- | --- | --- | --- | --- | --- | --- | --- | --- | --- | --- |
| | All | Old | New | All | Old | New | All | Old | New | All | Old | New | All | Old | New | All | Old | New |
| CLIP-H/14 | 98.5 | 98.3 | 98.6 | 86.4 | 86.2 | 86.9 | 78.3 | 78.5 | 78.2 | 89.2 | 93.1 | 87.3 | 95.7 | 95.1 | 96.0 | 93.5 | 93.3 | 93.9 |
| CLIP-B/16 | 98.2 | 98.4 | 98.0 | 85.7 | 85.1 | 86.6 | 77.2 | 76.8 | 77.7 | 87.8 | 91.2 | 85.5 | 93.8 | 94.3 | 93.4 | 92.0 | 92.3 | 91.8 |

Table E.8: Effect of the hyper-parameter $\alpha$ in RTA strategy.

| Data $\diagdown$ $\alpha$ | 0.2 | 0.3 | 0.4 | 0.5 | 0.6 | 0.7 | 0.8 |
| --- | --- | --- | --- | --- | --- | --- | --- |
| CIFAR-10 | 98.3 | 98.4 | 98.4 | **98.5** | 98.3 | 98.2 | 97.8 |
| CIFAR-100 | 86.0 | 86.6 | **86.7** | 86.4 | 85.8 | 85.2 | 84.1 |
| Stanford Cars | 87.1 | 88.7 | 88.9 | **89.2** | 88.8 | 87.2 | 86.4 |
| Flowers102 | 92.9 | 93.3 | 93.3 | **93.5** | 92.8 | 92.1 | 91.5 |

we can read that integrating more information from candidates yields consistent performance improvements. Specifically, the proposed RTA achieves its best performance when the number of candidates for tags and attributes is set to 3 or 4, respectively.

### E.3 EVALUATION ON REPRESENTATION LEARNING

**Evaluation on More Inter-Modal Alignment Methods.** To further evaluate the necessity of inter-modal alignment, we report the performance of using different inter-modal representation loss for training our framework. Specifically, we reproduce the cross-modal instance consistency objective (CICO) which is proposed in GET to serve as the inter-modal alignment constraint:

$$\mathcal{L}_{\text{CICO}} = \frac{1}{2|B|} \sum_{i \in B} \left( D_{\text{KL}}(\boldsymbol{s}_i^{\text{T}} \| \boldsymbol{s}_i^{\text{I}}) + D_{\text{KL}}(\boldsymbol{s}_i^{\text{I}} \| \boldsymbol{s}_i^{\text{T}}) \right), \tag{12}$$

where $D_{\text{KL}}$ is the Kullback-Leibler divergence, $B$ denotes the mini-batch data, $\boldsymbol{s}_i^{\text{I}} = \text{Softmax}(\boldsymbol{z}_i^{\text{I}^\top} \mathcal{A}^{\text{I}})$ and $\boldsymbol{s}_i^{\text{T}} = \text{Softmax}(\boldsymbol{z}_i^{\text{T}^\top} \mathcal{A}^{\text{T}})$ measure the distance between the $i$-th image/text embeddings and prototypes, and $\mathcal{A}^{\text{I}}, \mathcal{A}^{\text{T}}$ are the prototypes calculated using the labeled anchors for each modality.

In Table E.9, we report the performance of using $\mathcal{L}_{\text{CICO}}$ and its combination with the intra-modal alignment losses for representation learning, in which the results are marked in gray. As can be seen that, both inter-modal alignment losses $\mathcal{L}_{\text{CLIP}}$ and $\mathcal{L}_{\text{CICO}}$ fail to provide performance improvements when combined with the intra-modal alignment loss, further verifying the argument that the inter-modal alignment is not necessary.

Table E.9: Evaluation of different representation learning methods. Average ACC (%) on "All" categories is reported. "N/A" denotes using frozen CLIP.

| Rep. Losses | Inter | Intra | CIFAR-10 | CIFAR-100 | CUB | Cars | Pets | Flowers |
| --- | --- | --- | --- | --- | --- | --- | --- | --- |
| N/A | ✗ | ✗ | 97.9 | 84.1 | 74.5 | 86.0 | 91.9 | 87.4 |
| $\mathcal{L}_{\text{CLIP}}$ | ✓ | ✗ | 98.3 | 86.0 | 76.6 | 86.7 | 93.9 | 89.7 |
| $\mathcal{L}_{\text{CICO}}$ | ✓ | ✗ | 98.0 | 85.0 | 76.4 | 86.1 | 94.9 | 87.2 |
| $\mathcal{L}_{\text{con}}$ | ✗ | ✓ | 98.4 | **86.7** | 77.5 | 87.9 | 94.9 | 91.8 |
| $\mathcal{L}_{\text{SSR}^2}$ | ✗ | ✓ | **98.5** | 86.4 | **78.3** | **89.2** | **95.7** | **93.5** |
| $\mathcal{L}_{\text{CLIP}} + \mathcal{L}_{\text{con}}$ | ✓ | ✓ | 98.2 | 86.3 | 78.0 | 86.7 | 95.0 | 90.9 |
| $\mathcal{L}_{\text{CICO}} + \mathcal{L}_{\text{con}}$ | ✓ | ✓ | 98.4 | 85.9 | 76.8 | 87.0 | 94.4 | 88.6 |
| $\mathcal{L}_{\text{CLIP}} + \mathcal{L}_{\text{SSR}^2}$ | ✓ | ✓ | 98.3 | 86.1 | 77.2 | 88.1 | 95.0 | 92.9 |
| $\mathcal{L}_{\text{CICO}} + \mathcal{L}_{\text{SSR}^2}$ | ✓ | ✓ | 98.3 | 86.1 | 76.7 | 87.5 | 95.5 | 92.1 |

**Evaluation on Uni-Modal Representation Learning.** As an intra-modal alignment loss, the proposed $\mathcal{L}_{\text{SSR}^2}$ can also be used for uni-modal GCD counterparts. Specifically, for GCD and simGCD frameworks, we keep their pre-trained models and clustering algorithms and replace the supervised

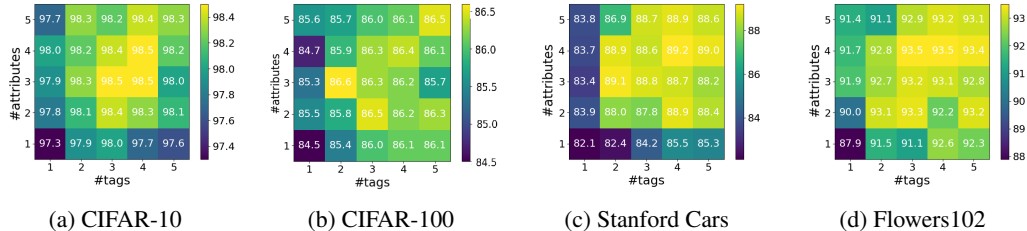

(a) CIFAR-10    (b) CIFAR-100    (c) Stanford Cars    (d) Flowers102

Figure E.1: Effect of the number of candidates.

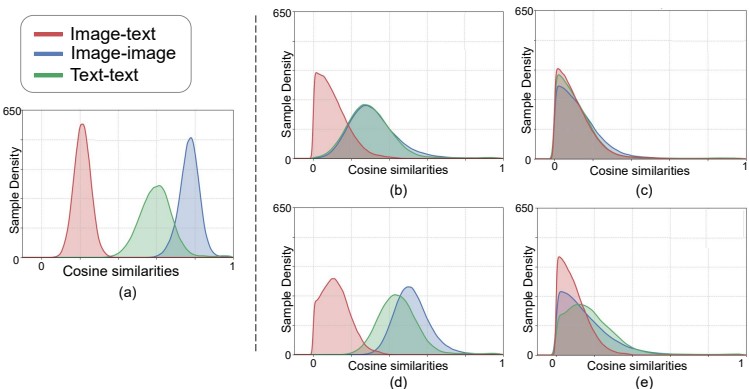

Figure E.2: Distribution of pairwise similarities on the Flowers102 dataset (a): at the beginning of training, (b)-(c): training with $\mathcal{L}_{\text{SSR}^2}$ at epochs 10 and 200, and (d)-(e): training with $\mathcal{L}_{\text{CLIP}}$ at epochs 10 and 200.

and unsupervised contrastive loss $\mathcal{L}_{\text{con}}$ with our proposed $\mathcal{L}_{\text{SSR}^2}$. As can be seen from Table E.10 that, using our $\mathcal{L}_{\text{SSR}^2}$ for representation learning achieves improvements on all tested datasets, while achieves a relatively less significant improvement on the CUB dataset.

**Evaluation on the Role of Inter-Modal and Intra-Modal Alignments.** To evaluate the role of inter-modal and intra-modal alignments in representation learning, we conduct experiments and display in Figure E.2 the distributions of the similarity computed by image-text (i.e., $\mathbf{Z}_{\text{T}}\mathbf{Z}_{\text{I}}^{\top}$), image-image (i.e., $\mathbf{Z}_{\text{I}}\mathbf{Z}_{\text{I}}^{\top}$), and text-text (i.e., $\mathbf{Z}_{\text{T}}\mathbf{Z}_{\text{T}}^{\top}$), obtained from our framework, when using the loss $\mathcal{L}_{\text{SSR}^2}$ or $\mathcal{L}_{\text{CLIP}}$ at the beginning of the training (i.e., epoch 0), the end of the warm-up (i.e., epoch 10) and the end of the training (i.e., epoch 200), respectively. Clearly, we observe from Figure E.2 that the distributions of inter-modal similarities and intra-modal similarities exhibit substantial divergence at the beginning, while the distributions of image-image and text-text similarities within each modality also differ notably. When using the proposed loss $\mathcal{L}_{\text{SSR}^2}$, the distributions of intra-modal similarities are well aligned in the warm-up stage, and all distributions gaps almost vanish at the end of the training. By contrast, using the loss $\mathcal{L}_{\text{CLIP}}$ can help to align the distributions of image-text similarities, whereas the image-image and text-text similarities, which are critical for GCD, remain poorly aligned.

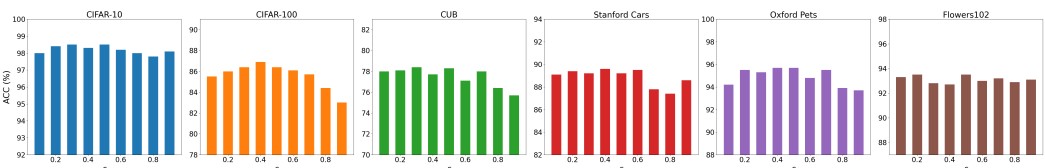

Figure E.3: ACC (%) with varying $\epsilon$ in SSR$^2$ across six benchmark datasets.

Table E.10: Comparison to uni-modal counterparts. Clustering ACC (%) on generic and fine-grained datasets.

| Method | CIFAR-10 | | | CIFAR-100 | | | CUB | | | Stanford Cars | | | Oxford Pets | | | Flowers102 | | |
|---|---|---|---|---|---|---|---|---|---|---|---|---|---|---|---|---|---|---|
| | All | Old | New | All | Old | New | All | Old | New | All | Old | New | All | Old | New | All | Old | New |
| GCD | 91.5 | **97.9** | 88.2 | 73.0 | 76.2 | 66.5 | 51.3 | 56.6 | 48.7 | 39.0 | 57.6 | 29.9 | 80.2 | 85.1 | 77.6 | 74.4 | 74.9 | 74.1 |
| GCD+SSR$^2$ | 92.5 | 96.4 | 91.6 | 73.9 | 79.0 | 63.2 | 51.9 | 55.0 | 47.1 | 47.9 | 56.1 | 47.3 | 83.6 | 87.7 | 79.8 | 80.0 | 83.3 | 78.5 |
| SimGCD | 97.1 | 95.1 | **98.1** | 80.1 | 81.2 | 77.8 | 60.3 | **65.6** | 57.7 | 53.8 | **71.9** | 45.0 | 87.7 | 85.9 | 88.6 | 71.3 | 80.9 | 66.5 |
| SimGCD+SSR$^2$ | **97.6** | 97.5 | 97.7 | **81.1** | **82.5** | **78.9** | **60.8** | 64.7 | **59.0** | **57.1** | 66.8 | **53.9** | **90.0** | **89.8** | **91.2** | **81.6** | **83.5** | **80.1** |

To further evaluate the role of inter-modal and intra-modal interactions, we combine our proposed intra-modal alignment loss $\mathcal{L}_{\mathrm{SSR}^2}$ with the inter-modal alignment loss $\mathcal{L}_{\mathrm{CLIP}}$ by:

$$\mathcal{L}_{\mathrm{SSR}^2} + \varphi \cdot \mathcal{L}_{\mathrm{CLIP}},$$

where $\varphi$ is the penalty weight of $\mathcal{L}_{\mathrm{CLIP}}$. In Figure E.4, we report the accuracy with varying penalty weight $\varphi$ on the Flowers102 dataset. Existing multi-modal GCD frameworks, such as GET, assume that the learning of inter-modal alignment does not affect that of intra-modal alignment, and naively treat these two learning processes in representation learning as both independent and equally important. However, we can see from Figure E.4 that the learning of inter-modal alignment significantly impairs the learning of intra-modal alignment as $\varphi$ increases.

**More Results on the Ratio of Edges.** Recall that we define the ratio of edges $R_e$ in our manuscript to quantize the intra-modal consistency of embeddings and explain why inter-modal alignment can be unnecessary. In this paragraph, we additionally report $R_e$ of more inter-modal alignment loss (i.e., $\mathcal{L}_{\mathrm{CLIP}}$), intra-modal alignment losses (i.e., $\mathcal{L}_{\mathrm{SSR}^2}$ and $\mathcal{L}_{\mathrm{con}}$) and their combinations (i.e., $\mathcal{L}_{\mathrm{SSR}^2} + \mathcal{L}_{\mathrm{CLIP}}$ and $\mathcal{L}_{\mathrm{con}} + \mathcal{L}_{\mathrm{CLIP}}$) on the Stanford Cars dataset.

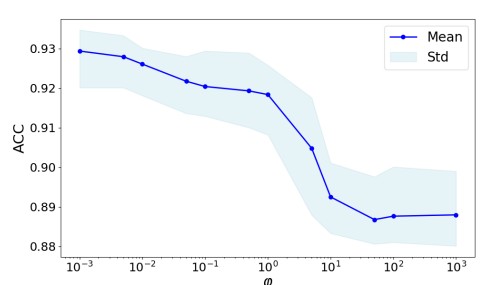

Figure E.4: Mean and standard accuracy (3 trials) of "All" categories with varying penalty weight $\varphi$ on Flowers102.

Furthermore, as shown in Figure E.5, the inter-modal alignment also damages the learning of other intra-modal losses such as the widely adopted supervised and unsupervised contrastive loss $\mathcal{L}_{\mathrm{con}}$. This confirms that the aforementioned trend is not exclusive to scenarios where $\mathcal{L}_{\mathrm{CLIP}}$ is combined with our proposed $\mathcal{L}_{\mathrm{SSR}^2}$.

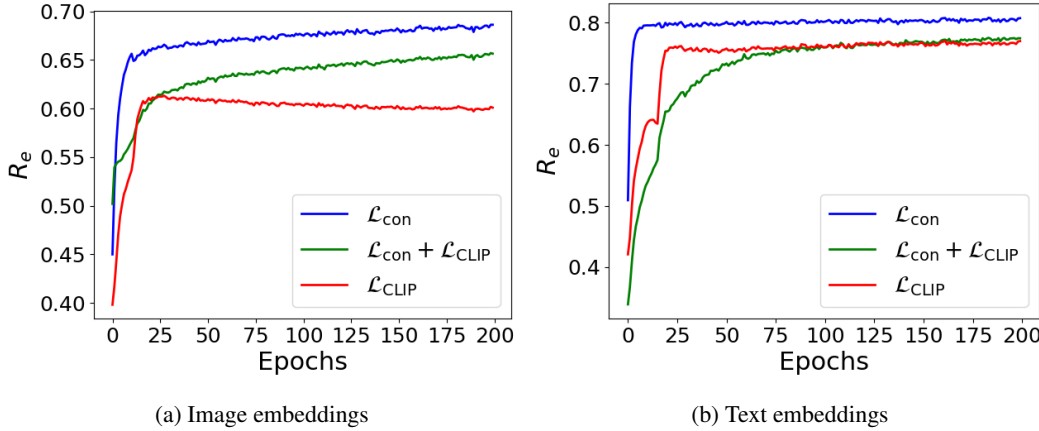

(a) Image embeddings      (b) Text embeddings

Figure E.5: $R_e$ curves with different losses on Stanford Cars.

### E.4 EVALUATION ON THE HYPER-PARAMETERS

We evaluate the impact of different hyper-parameters introduced by our representation learning method. Recall that the scaling parameter $\epsilon > 0$ is specific to our method, while the settings of all other hyper-parameters follow those of existing GCD counterparts. In Figure E.3, we report the clustering accuracy with varying value of $\epsilon$ on six benchmark datasets, i.e., CIFAR-10, CIFAR-100, CUB, Stanford Cars, Oxford Pets and Flowers102. As can be seen, our framework is not sensitive to $\epsilon$ and achieves the best performance when $\epsilon$ is in the range of $[0.2, 0.5]$.

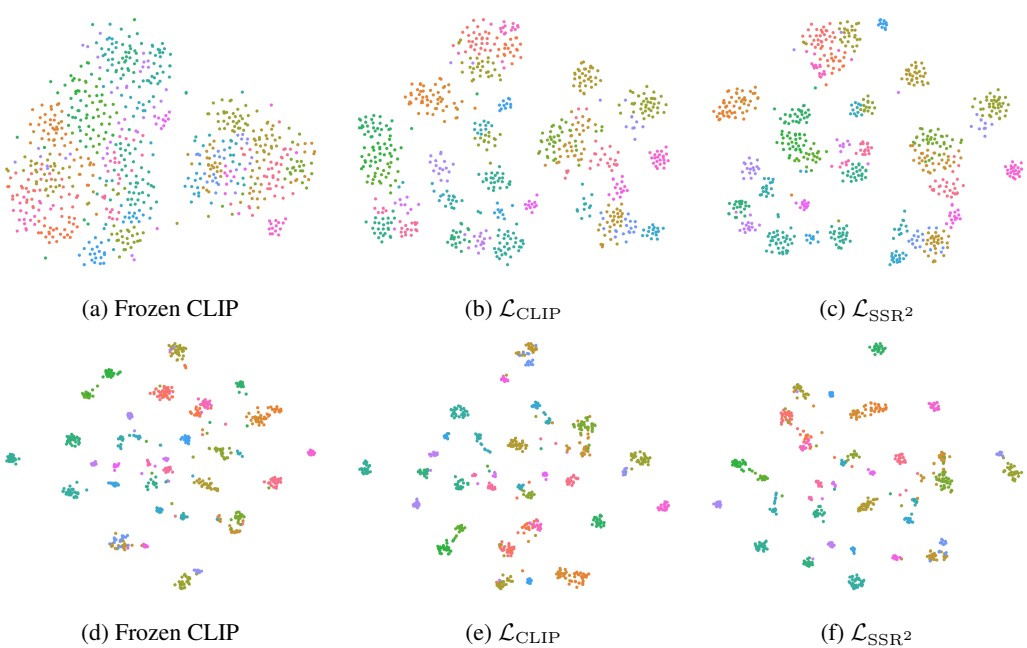

| (a) Frozen CLIP | (b) $\mathcal{L}_{\mathrm{CLIP}}$ | (c) $\mathcal{L}_{\mathrm{SSR}^2}$ |
|---|---|---|
| (d) Frozen CLIP | (e) $\mathcal{L}_{\mathrm{CLIP}}$ | (f) $\mathcal{L}_{\mathrm{SSR}^2}$ |

Figure E.6: Visualization of image embeddings (**top**) and text embeddings (**bottom**) with varying representation learning methods on the Oxford Pets dataset.

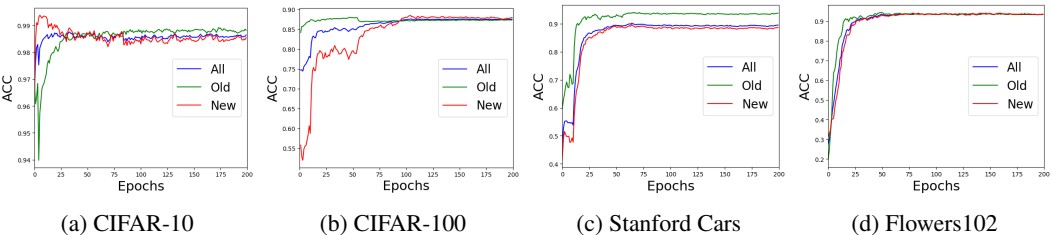

| (a) CIFAR-10 | (b) CIFAR-100 | (c) Stanford Cars | (d) Flowers102 |
|---|---|---|---|

Figure E.7: ACC curves on benchmark datasets.

### E.5 VISUALIZATION

In Figure E.6, we visualize the image embeddings and text embeddings of our framework using different representation learning methods on the Oxford Pet dataset. As can be seen, the distribution of image embeddings generated by the frozen CLIP image encoder is rather diffuse and can be roughly divided into two hyper-classes ("cat" and "dog"). In contrast, thanks to our proposed Retrieval-based Text Aggregation (RTA) strategy, the distribution of text embeddings produced by the frozen CLIP text encoder exhibits significantly greater discriminability. Such discrepancies between the two modalities may explain why optimizing the inter-modal alignment loss can enhance intra-class discriminability to some extent—the learning of image embeddings is largely guided by the inherently discriminative text embeddings. Meanwhile, the gap between the two hyper-class still

remains (See, e.g., Figure E.6a and Figure E.6b for comparison). In contrast, our method learns discriminative and well-balanced representations for both modalities.

## E.6 LEARNING CURVES

We plot the learning curves with respect to the clustering accuracy of our method on CIFAR-100, CIFAR-100, Stanford Cars and Flowers102 in Figure E.7. We can observe that our SSR$^2$-GCD converges and achieves stable clustering results on "All" categories within 50 epochs.

