# OpenReview forum: "Multi-Modal Representation Learning via Semi-Supervised Rate Reduction for Generalized Category Discovery"
_ICLR.cc/2026/Conference — ICLR 2026 Conference Withdrawn Submission_

### Official Review · Reviewer_zvop · 2025-10-29

**Soundness:** 2
**Presentation:** 3
**Contribution:** 1
**Rating:** 2
**Confidence:** 5

**Summary:**

This paper proposes SSR²-GCD, a multi-modal framework for Generalized Category Discovery (GCD) that integrates semi-supervised rate reduction and retrieval-based text aggregation, while challenging the necessity of inter-modal alignment. Despite addressing a relevant problem in open-set recognition, the work suffers from critical gaps in methodological rigor, experimental completeness, theoretical justification, and comparative analysis. Overall, I vote for rejection.

**Strengths:**

This paper proposes SSR²-GCD, a multi-modal framework for Generalized Category Discovery (GCD) that integrates semi-supervised rate reduction and retrieval-based text aggregation, while challenging the necessity of inter-modal alignment.

**Weaknesses:**

1. Insufficient Methodological Rigor and Clarity. The core SSR² principle is claimed to enable "even compression" of known and unknown category embeddings, but its adaptation from the supervised Maximal Coding Rate Reduction (MCRR) (Yu et al., 2020) to semi-supervised GCD is poorly justified and documented:
2. The paper’s central claim that "inter-modal alignment is non-essential" is supported only by experimental correlation, but lacks causal or theoretical explanation
3. It does not address why inter-modal alignment (critical for other multi-modal tasks like zero-shot recognition) fails here. For example, is the failure due to GCD’s focus on intra-modal clustering, or due to the RTA strategy already capturing cross-modal consistency? No analysis of cross-modal embedding overlap (e.g., via mutual information) is provided.
4. The RTA strategy (aggregating top-c tags/attributes) is underdeveloped and lacks motivation
5. The experimental setup fails to contextualize SSR²-GCD against state-of-the-art (SOTA) and edge cases, limiting generalizability
6. Missing critical baselines. The paper compares two multi-modal methods, such as TextGCD and GET, but overlooks recent GCD works that address similar challenges
7. The paper does not report computational costs (e.g., training time, GPU memory) or inference latency—key metrics for multi-modal methods. For example, RTA’s aggregation of top-4 candidates may increase text encoder compute, but no comparison to TextGCD (which uses top-3 tags/top-2 attributes) is provided to assess efficiency trade-offs.

**Questions:**

See the above comments

---

### Official Review · Reviewer_17T9 · 2025-10-31

**Soundness:** 3
**Presentation:** 2
**Contribution:** 3
**Rating:** 6
**Confidence:** 4

**Summary:**

This paper proposes SSR2-GCD: semi-supervised rate reduction + retrieval aggregated texts + dual classifiers for generalized category discovery. The authors show that SSR2 yields better-balanced representations and improved ACC on multiple benchmarks. They argue inter-modal contrastive alignment is not required and may be harmful.

**Strengths:**

- The paper smartly extends rate-reduction to a semi-supervised multi-modal setup, combining global, supervised, and unsupervised terms to balance modalities and leverage pseudo-labels.

- Re curves and effective rank analyses clearly show that SSR2 yields more balanced and diverse representations, validating its theoretical goal of even compression and offering interpretable insight into representation behavior.

- The paper reports consistent improvements on diverse benchmarks, with ablation studies that convincingly isolate and confirm the contribution of each key module.

**Weaknesses:**

- Some claims about “inter-modal alignment being unnecessary” are empirical and might need more nuance (depends on text quality / domain).

- No results on ImageNet-100/1K (listed in Table D.1 but absent in main tables).

- In Equations (2) and (3), as well as in the subsequent textual explanations, inconsistencies are evident in symbol usage and formulation. For instance, the equations appear to omit the term  . The authors are advised to perform a thorough review.

**Questions:**

- Could the authors comment on whether RTA would still help if the prompt lexicon is noisy or small?

- Are all CLIP text embeddings computed online per batch or cached per prompt candidate?

- How often are the pseudo-labels Y updated during training? Are they from soft or hard argmax?

---

### Official Review · Reviewer_93Qu · 2025-11-01

**Soundness:** 2
**Presentation:** 3
**Contribution:** 2
**Rating:** 4
**Confidence:** 3

**Summary:**

This paper introduces a novel method for multimodal Generalized Category Discovery (GCD) based on CLIP. The proposed method SSR$^2$-GCD is an end-to-end learning framework with the core component of Semi-supervised Rate Reduction (SSR$^2$). Specifically, it encourages both the labeled and unlabeled data to learn compact intra-class features and drives the entire dataset to be distributed more uniformly. The paper also improves detailed strategies including retrieval-based text aggregation (RTA) and dual-branch clustering. The proposed framework achieves state-of-the-art performance on multiple GCD benchmarks. Analytical experiments confirm that all the components achieve expected effects and make individual contributions.

**Strengths:**

1. The paper is clearly written.
2. The proposed SSR$^2$ method is well motivated and sounds reasonable.
3. The whole framework achieves state-of-the-art performance with large improvements compared to baselines.
4. The contribution of each component is validated in analyses.

**Weaknesses:**

1. The employed Maximal Coding Rate Reduction is not original. Although it’s designed in the context of general supervised learning, applying it to GCD is mostly straightforward. The only adaptation is to use pseudo labels for the unlabeled subset.
2. Dual-branch clustering and RTA also make considerable contributions in the whole framework (according to the ablation study), but are incremental updates without much innovation.
3. Code is not provided.
4. The results in Figure 3 are not quite informative to me. Since $L_{con}$ contains a supervised contrastive loss on the known classes, it’s not surprising that the known classes have decreased feature rank. SSR$^2$ maintains similar feature ranks for known and unknown classes since it doesn’t include supervised signals. My understanding is that the SSR$^2$ is similar to traditional clustering as they both encourage compact distribution in unsupervised manners, so it’s better to compare its mechanism and effects with the clustering methods.
5. Some implementation and experimental details are not clear (see Questions).

**Questions:**

1. The SSR$^2$ illustration in Figure 1 is not very intuitive. Can you explain why the data points lie on two lines?
2. In line 195, should $\hat{y}_i$ be $y_i$?
3. What’s the specific reason for the design of Eq (6)? Why do you use a constant weight for all $i>1$ rather than smooth weights?
4. Regarding $R^s_c$ in Eq (7), should the number of categories be smaller than $k$?
5. Why do you use Diag($Y_j$) in the Maximal Coding Rate Reduction? The labels are categorial variables and don’t have numerical meanings. I guess it should be a kind of binary masking which is used to select the data belonging to a specific class.
6. Did you directly reuse the numbers reported in their papers? If so, did you use the same category and subset split to create the labeled and unlabeled dataset? That’s essential for fair comparison.
7. Can you show some examples of the generated tags and attributes?
8. Did you quantify the contribution of the image and text SSR$^2$ defined in Eq (8)?

---

### Note · Authors · 2025-11-14

I have read and agree with the venue's withdrawal policy on behalf of myself and my co-authors.